# Double-Chain Cationic Surfactants: Swelling, Structure, Phase Transitions and Additive Effects

**DOI:** 10.3390/molecules26133946

**Published:** 2021-06-28

**Authors:** Rui A. Gonçalves, Yeng-Ming Lam, Björn Lindman

**Affiliations:** 1School of Materials Science and Engineering, Nanyang Technological University, Singapore 639798, Singapore; rui.goncalves@ntu.edu.sg; 2Centre for Chemistry and Chemical Engineering, Department of Chemistry, Division of Physical Chemistry, Lund University, SE-221 00 Lund, Sweden; 3Coimbra Chemistry Center (CQC), Faculty of Sciences and Technology, University of Coimbra, 3004-535 Coimbra, Portugal

**Keywords:** double-chain surfactant, lamellar gel, lamellar liquid crystal, hair conditioner, surfactant packing, additive effects

## Abstract

Double-chain amphiphilic compounds, including surfactants and lipids, have broad significance in applications like personal care and biology. A study on the phase structures and their transitions focusing on dioctadecyldimethylammonium chloride (DODAC), used inter alia in hair conditioners, is presented. The phase behaviour is dominated by two bilayer lamellar phases, L_β_ and L_α_, with “solid” and “melted” alkyl chains, respectively. In particular, the study is focused on the effect of additives of different polarity on the phase transitions and structures. The main techniques used for investigation were differential scanning calorimetry (DSC) and small- and wide-angle X-ray scattering (SAXS and WAXS). From the WAXS reflections, the distance between the alkyl chains in the bilayers was obtained, and from SAXS, the thicknesses of the surfactant and water layers. The L_α_ phase was found to have a bilayer structure, generally found for most surfactants; a L_β_ phase made up of bilayers with considerable chain tilting and interdigitation was also identified. Depending mainly on the polarity of the additives, their effects on the phase stabilities and structure vary. Compounds like urea have no significant effect, while fatty acids and fatty alcohols have significant effects, but which are quite different depending on the nonpolar part. In most cases, L_β_ and L_α_ phases exist over wide composition ranges; certain additives induce transitions to other phases, which include cubic, reversed hexagonal liquid crystals and bicontinuous liquid phases. For a system containing additives, which induce a significant lowering of the L_β_–L_α_ transition, we identified the possibility of a triggered phase transition via dilution with water.

## 1. Introduction

Surfactant formulations are widely used in many areas of industry and daily life. They are generally complex, and contain many components. Additive effects in surfactant systems play a central role in formulation science and technology, and fundamental research in this area aims to understand the underlying interactions and mechanisms. Additives can increase solubility and modulate association. These additives tend to interact differently with water-soluble and water-insoluble surfactants.

When it was observed, ca. 100 years ago, that physicochemical parameters of aqueous surfactant solutions display a break at a certain concentration [1,2,3], this not only meant a new understanding of the association behaviour (micelle formation), but also provided an excellent handle to characterise and compare different surfactants. The critical micelle concentration, CMC, has continued to be the single most-studied aspect of surfactants, and extensive tables were already published long ago [4].

While the notion of large aggregates, micelles, became established at an early stage, several molecular aspects of micelles remained controversial for a long time. In his pioneering work, Hartley [5] proposed that polar surfactants form spherical micelles characterised by high molecular mobility and disorder, with an interior devoid of water. The Hartley micelle took a long time to be generally accepted, and alternative structures were discussed well into the 1980s. However, with NMR and other techniques, it could be firmly established that firstly there is no alkyl chain-water contact, except at the micelle surface, and that the alkyl chain mobility is close to that of liquid hydrocarbons [6,7,8,9], thus leading to a consensus on the Hartley micelle [10,11].

For single-chain ionic surfactants, a rather simple picture appears regarding the effects of additives on self-assembly, as can be inferred from changes in the CMC, and micellar growth from spherical to worm-like micelles. Highly water-soluble compounds weaken the self-assembly, as exemplified by urea, glycol, dioxane etc., easily understood in terms of a weakening of hydrophobic interactions (reviewed in [12]). Electrolyte addition strengthens self-assembly due to simple electrostatic effects; calculations using the Poisson-Boltzmann equation can quantitatively describe changes in the CMC and phase diagrams [13,14].

All other types of additives will promote self-assembly to various degrees, depending on the polarity and polarisability of the cosolute. Alcohols induce a lowering of the CMC, and promote micellar growth more so than the larger the hydrophobic part of the alcohol; these systems can be effectively described in terms of mixed micelle formation, taking the alcohol solubility as its CMC. Aromatic compounds strengthen self-assembly by locating at the micelle surface. Alkanes have a minor influence on the CMC, and are located in the interior of the micelles.

Double-chain ionic surfactants are generally not soluble in water, and experimental techniques used for studying single-chain ones are, in general, not applicable. These surfactants have no CMC, and instead the most important piece of information lies in the phase diagrams. One very significant consideration for such surfactant formulation is the stability, and thus formulation work must be based on phase diagrams. It is no surprise, therefore, that pioneering work on surfactant phase diagrams came from industrial laboratories such as Procter & Gamble, Unilever, Kao and Henkel etc., or academic researchers that were supported by the industry, whose leading workers include Shinoda, Ekwall, Tiddy, Laughlin and Schwuger, to name a few [15,16,17,18,19,20].

While extensive attention was given to cosolute effects for single-chain surfactants at an early stage, far less attention has been devoted to additive effects for the self-assembly of water-insoluble ionic surfactants. These surfactants do not form micelles, but typically self-assemble into structures with infinite bilayers. The present study is focused on swelling and additive effects on a double-chain cationic surfactant system, dioctadecyldimethylammonium chloride (DODAC), a representative hydrophobic surfactant; as a comparison, the swelling of dioctadecyldimethylammonium bromide (DODAB) is also investigated and presented in this study. In addition to water, such surfactants form bilayer structures in essentially two different phases, one liquid crystalline phase, often denoted L_α_, and one lamellar gel phase, L_β_, characterised by rigid alkyl chains rather than liquid as in L_α_. In this paper, we present studies of the swelling behaviour and phase transitions in the DODAC-water system, and in particular the effect of additives on phase stability and structure. In a previous study [21], we already presented limited results pertaining to a few additives, but here we include a much larger number of additives, as well as present other types of observations. From a mechanistic point of view, it is of considerable interest to compare the observations with analogous ones for single-chain water-soluble, i.e., micellar, surfactants.

Major uses of these surfactants investigated are as hair and textile conditioners/softeners, but they also have many other applications. Compounds like DODAC function well for these purposes. The conditioning action is due to the L_β_ phase, which is a stable phase at room temperature. It is normally formulated as kinetically stable dispersions of L_β_ in water. At higher temperatures, the thermodynamically stable phase is L_α_; from a stability perspective, it would be preferable for the formulation to be a stable system rather than a dispersion. As we have already alluded to in our previous report [22], a further aim of our work was to investigate the possibility of designing formulations that contain the L_α_ phase, but spontaneously transform into L_β_ on dilution with water, mimicking the situation in practical use. Thus, we were interested in adding substances that appreciably lower the phase transition temperature, but are soluble in water so that upon dilution, the L_β_ phase forms spontaneously.

## 2. Results and Discussion

### 2.1. Thermal Behaviour of DODAC and DODAB

DODAC and DODAB are highly insoluble in water (10^−5^–10^−10^ M) [23], and according to pure geometrical constraints (cf. above), they spontaneously self-assemble into lamellar structures [24]. DODAC was found to form more stable single phases in water as compared to DODAB; it can be seen that there is the presence of small crystals in a vial containing DODAB in water – crystals that were not observed for DODAC solutions below the Krafft temperature (Appendix A). Initial observations by the naked eye and with polarised optical microscopy (POM) gave qualitative information on dispersibility and the presence of lamellar phases. The inspection of the POM micrographs of DODAB shows that anisotropic structures are formed, but it was not a one-phase system (Appendix A). Dispersions in two-phase regions were outside the scope of our work, but it was inferred that DODAB is more prone than DODAC to being dispersed. The work focused instead on phase transitions and phase structures in the DODAC-water system, with special attention on the effect of additives. Limited studies on the DODAB-water system were performed for comparison.

In order to examine the phase behaviour of the surfactant in water, differential scanning calorimetry (DSC) measurements were conducted. The DSC experiment tracked thermal transitions and their associated thermodynamic properties, as well as the transition temperature (T_m_) and enthalpy (ΔH). We have in some detail investigated by DSC the transition between the L_β_ and L_α_ phases, including the effect of scan rate, reversibility, effect of sample history, and in particular the composition. A summary of the results was reported previously [21], and more details can be found in Appendix A.

Important for the present study is that for the DODAC-water system, we found a well-defined transition temperature (ca. 45 °C) and an enthalpy change (ca. 44 kJ/mol) that do not significantly change as the sample composition is varied over wide ranges (Table 1, Figure 1a,b). For DODAB, both the transition temperature (ca. 53 °C) and the enthalpy change (ca. 100 kJ/mol) are higher, but again with no significant change with composition over wide ranges (Table 1, Figure 1c,d). For DODAB, the behaviour was less clear-cut and included pre-transition and hysteresis effects. These results are in agreement with the studies previously reported by Kodama et al. for a series of surfactants in water, including DODAB [25,26]. These authors demonstrated that a gel phase exists in both stable and metastable states, and that it could be tuned by cooling the sample to -20 °C, where the metastable gel phase was suppressed. Previous work on a shorter homologue of DODAB has also indicated complex behaviour, including the existence of two lamellar liquid crystalline phases [27].

As described, DODAB in water shows a higher T_m_ and associated ΔH_m_ when compared to DODAC. This indicates a more stable L_β_ phase in the case of DODAB. We attribute the stronger interaction of Br-ions with the surfactant head-groups to their larger polarisability. An increased counterion concentration in the interfacial region reduces the electrostatic repulsion. Therefore, the surfactant molecules pack closer to each other, which results in higher melting temperatures and requires a larger amount of energy for the melting transformation to occur. Analogous differences between the two counterions are well documented for single-chain surfactants; examples include micellar growth, phase diagrams and Krafft points [9,28], as well as electrolyte effects on the solubility of nonionic surfactants and polymers in water [29].

The observations for DODAC that the transition temperature and enthalpy do not change with composition indicate that we deal with the same transition, L_β_ to L_α_, over the entire composition range. As we will see, this is confirmed by X-ray diffraction studies.

### 2.2. SAXS and WAXS Studies

#### 2.2.1. Swelling of DODAC in Water

Scans of DODAC-water samples using SAXS and WAXS over a wide temperature range were performed to investigate the self-assembled structure of the surfactant during the thermal transitions observed by DSC. The T_m_ of DODAC in water was determined to be 45.1 °C, and simultaneous SAXS and WAXS data from 35 to 50 °C and back down to 35 °C using a 1 °C scan step were collected. The study aimed to elucidate the nature of the phase transitions that occur in DODAC solutions.

The presence of lamellar phases is demonstrated by the 1:2:3… pattern of diffraction peaks in the SAXS results. The two lamellar phases are easily distinguished by the sharp diffraction peaks in the WAXS patterns for L_β_, in contrast to a broad peak for the L_α_ phase. At 35 °C, the SAXS pattern for 35 wt% DODAC shows a lamellar phase with a typical spacing of 69 Å (Figure 2a). The WAXS pattern displays a single and sharp reflection at q = 1.505 Å^−1^, which corresponds to a distance of 4.2 Å (Figure 2b). Therefore, these results indicate that DODAC self-assembles into a lamellar phase with alkyl chains in a “frozen” state below the T_m_. Under heating, the development of two coexisting lamellar structures was observed, starting from 40 °C (Figure 2d) — one with interlamellar spacing (d_sp_) of 78 Å and a second with a d_sp_ of 69 Å. The peak corresponding to the thicker lamellar structure gradually became more intense, until a temperature above the T_m_ was reached. With a further increase in temperature, only one lamellar structure was identified at 47 °C with a d_sp_ of 84 Å. A disordered fluid state of the alkyl chains of DODAC was identified by the disappearance of the sharp reflection in the wide-angle regime, and the development of a broad peak slightly shifted to lower q values during the heating scan (Figure 2g). These scattering patterns indicate a transition of chains from a highly ordered phase, with a typical chain-to-chain spacing of 4.1–4.2 Å, to a fluid phase, liquid crystalline, with a 4.5–4.6 Å distance between alkyl chains, as schematically depicted in Figure 2c. Therefore, small-angle X-ray scattering patterns of the lamellar structures of the L_β_ and L_α_ phases can be confirmed, and are in agreement with what was observed generally for these phases [30].

As argued above, there is a clear segregation into polar and non-polar regions for micelle-forming surfactants. Due to lower polarity and concomitant closer packing, the tendency of such segregation would be even more pronounced in lamellar phases of double-chain surfactants. This means that for the analysis of the swelling behaviour, a simple picture like that shown in Figure 2c is expected to offer a good basis. To determine parameters such as the surfactant bilayer thickness (d_bi_), the water layer thickness (d_w_) and the area per surfactant molecule (a), an analysis of the scattering patterns was carried out. As shown in Figure 2c, the surfactant system was assumed to comprise two regions, the bilayer — containing the surfactant hydrocarbon chains — and the water layer — containing the water — as well as the surfactant head-groups and the counterions.

For a liquid system, like the L_α_ phase, the alkyl chains will be disordered and have a larger cross-section area than in the L_β_ phase. This presumption is directly confirmed from the alkyl chain distances inferred from WAXS diffraction peaks.

With a larger cross-section area of the alkyl chains in the L_α_ phase, for a given surfactant volume fraction we would then expect thinner water layers, and thus shorter spacings in the L_α_ phase than in L_β_. However, over the entire composition range, this is opposite to observations; this means that the organisation in the L_β_ phase cannot follow the idealised picture of Figure 2c.

In literature, it is possible to find a number of suggested deviations from the ideal behaviour of Figure 2c, namely rippled (P_β_), tilted (L_β_’) and interdigitated (L_β_int) gel phases [23,31,32,33]; see Figure 3. By investigating the changes in the SAXS spacing data on dilution with water, some insight into the structure can be obtained.

The swelling behaviour of DODAC in water, especially for surfactant concentrations that display a single and stable phase (>33 wt% [34]) down to a rather low surfactant concentration of 5 wt%, was investigated. The small and wide-angle X-ray scattering patterns collected at one temperature below and one above the T_m_, i.e., at 25 °C and 50 °C, are presented in Figure 4.

For both L_β_ and L_α_, we found a shift of the first scattered peak towards lower q values with increasing amounts of water in the system, which suggests a progressive interlamellar swelling with the addition of water. With increasing amounts of water between the bilayers, the sharp reflection peak of the hydrocarbon chains in the “solid-state” in the wide-angle became progressively less intense, which is expected with the increase of the water layer thickness. Using Equations (1)–(3), the values of the interlamellar spacing (d_sp_), bilayer thickness (d_bi_), polar layer thickness (d_w_) and area per surfactant molecule (a) of DODAC bilayers in water at temperatures below (25 °C) and above (50 °C) the main phase transition temperature (T_m_) were calculated; they are summarised in Table 2.

Structural parameters of the lamellar phase, such as the bilayer volume fraction (Φ_bi_), the area per surfactant molecule (a) and the bilayer thickness (d_bi_), were calculated from the SAXS and WAXS results, using Equation (1) and Equation (2) [35]
(1)$$d_{\mathrm{sp}}=\frac{2\pi }{q}$$
(2)$$q=\frac{2\pi }{d_{\mathrm{sp}}}=2\pi \frac{\Phi_{\mathrm{bi}}}{d_{\mathrm{bi}}}=\pi \frac{\Phi_{\mathrm{bi}}a}{v_{\mathrm{bi}}}$$
where q is the first-order Bragg peak; Φ_bi_ is the volume fraction of the total hydrophobic constituents and $v_{\mathrm{bi}}$ is the molar volume of the total amphiphiles present in the bilayer.

The non-polar domain only contains the hydrophobic segments of the surfactant; therefore, the Φ_bi_ can be calculated using Equation (3) [35]
(3)$$\Phi_{\mathrm{bi}}=\frac{v_{\mathrm{hc},S}\times \frac{w_{S}}{M_{S}}}{v_{S}\times \frac{w_{S}}{M_{S}}+v_{W}\times \frac{w_{W}}{M_{W}}}$$ where w is the weight fraction, and M is the molar mass of the various components denoted as surfactant (S) and water (W); and v is the molecular volume given by the sum of the partial molar volume of the hydrocarbon chain (hc) and the polar head-group of the molecule. To analyse the SAXS and WAXS results, the values used for the various parameters were $v_{\mathrm{hc},S}$ = 1026 Å^3^ [36], $v_{S}$ = 1126 Å^3^ [37] and $v_{W}$ = 30 Å^3^ [35]. Since the hydrated radius of bromide and chloride was the same, at 3.3 Å [38], the volume of DODAB and DODAC was considered to be similar.

Figure 4g shows the swelling behaviour of DODAC in water. The slope of the straight line of the swelling in the gel phase (left panel, squares) is 0.97, while it is 1.00 for the liquid crystalline phase (right panel, circles). The value of the slope of the two straight lines suggests that a one-dimensional swelling occurs because the interlamellar spacing is inversely proportional to the bilayer volume fraction; this indicates that the surfactant bilayers remain unaffected in the swelling process. The value of d_sp_ increases with the water content for both phases corresponding to an increased thickness of the water layers. This observation is generally expected for all surfactants, with the difference being that nonionic surfactants normally do not swell much. A large water uptake and swelling can be understood from the electrostatic repulsions, mainly the counterion entropy [23,39]. Since there is a progressive swelling with the water content over the entire range, only one phase, lamellar, is indicated; no coexisting water phase, either below or above the T_m_, is found. These findings point to a significant swelling of the DODAC in water in the gel phase, and not only in the liquid crystalline phase.

As indicated above, the liquid crystalline phase yields larger d_sp_ values (Table 2 and Figure 4g). This is a result of the surfactant alkyl chains changing conformation from an all-trans state to a disordered one upon melting, causing an effect of a shorter end-to-end distance and an overall increase in the effective cross-sectional area. Supposing the chains adopt a perpendicular conformation to the interface, it is expected that higher d_sp_ values would occur for the L_β_ than the L_α_ phase, which could be observed by a potential shift to higher Bragg reflection q values for the L_α_ phase. Our results show the opposite effect, however, implying that the L_β_ phase is either orientated in a different direction or has an alternative packing arrangement. Calculating the average bilayer thickness supports the shift to a L_α_ phase from an all-trans conformation, whereby all the chains are perpendicular to the interface, as a thickness of 31 Å at 50 °C is in line with an expected 30% decrease in thickness upon a “solid-like” to “fluid-like” chain transformation [38]. This suggests an interdigitation or tilting of the hydrocarbon chains in the L_β_ phase, or a combination of these. The phenomena described currently have multiple explanations, and further research is needed. However, we expect that mismatches between cross-sectional areas of the chain and head-groups could lead to variations in the critical packing parameter, which is of importance to the stability of lamellar phases [40].

If the effect was ascribed to tilting alone, the tilting angle of a hydrocarbon chain must be approximately 59°, which seems to be an extremely large angle. Therefore, the existence of a tilted and interdigitated structure in the gel phase is proposed. An alkyl chain with 18 carbons, fully stretched, perpendicular to the interface and in the “solid-like” state, was expected to measure around 24 Å in length [41].

The experimental results, displayed in Table 2, show that the bilayer of the gel phase was thinner than the bilayer of the liquid crystalline phase. Figure 4h shows the lamellar swelling evolution for DODAC in water. In this graph, the experimentally determined d_sp_ of the L_β_ and L_α_ phases are compared with a theoretical maximum interlamellar spacing prediction (d_max_), using Equation (4) [42]
(4)$$d_{\max}\;\left(Å\right)=\frac{\sqrt{3}\times 10^{27}}{a^{2}\mathrm{CR}}$$ where a is the chain-to-chain distance determined using wide-angle X-ray scattering as 4.2 Å and 4.6 Å for the L_β_ and L_α_ phase, respectively [21,23,42,43]. C is the surfactant concentration (mol/dm^3^), and R is the Avogadro’s constant.

By comparing the calculated d_max_ and the measured d_sp_, two conclusions can be made. Firstly, the lamellar packing was not destroyed with the addition of water. This evidence suggests that the system was capable of taking up large amounts of water, corresponding to a progressive increase of the water layer thickness. In addition, there was no significant effect on the T_m_ and the enthalpy of phase transition (ΔH_m_) with increasing amounts of water, which supported the idea of the maintenance of the lamellar packing. Secondly, the measured d_sp_ values of the L_α_ phase were always larger than the L_β_ phase. However, the opposite behaviour was predicted, which suggests an alternative chain packing in the L_β_ phase. Regardless of the amount of water present in the system, the same trend was observed for all the samples. Therefore, a tilted and/or interdigitated structure in the gel phase was suggested.

#### 2.2.2. Swelling of DODAB in Water

SAXS and WAXS data was also collected for comparison for the DODAB-water system. Here a more complex behaviour was observed, with additional transitions and metastable states. A pre-transition observed under heating did not lead to a different lamellar structure. Thus, SAXS patterns below and above the pre-transition temperature overlapped, and displayed a lamellar structure with the same d_sp_ value (Figure 5).

Above the Krafft temperature, both surfactants display a large lamellar region, with DODAB developing two different lamellar mesophases at different DODAB concentrations [34,44]. Our results showed various new observations that can complement the previous phase studies conducted on these surfactants. A comparison of the deduced geometrical characterisation of the two surfactants is given in Table 3. Similar to DODAC, DODAB provides wide ranges of concentration and temperature in the L_α_ and L_β_ phases; in a temperature range around the transition temperature, a coexistence of the two phases was clearly shown in the diffraction patterns (Figure 5a). An example is given in Figure 6, displaying a striking difference in structure between the two phases. A lamellar gel structure at low temperatures is identified with a d_sp_ of 36 Å. The DODAB hydrocarbon chains are composed of 18 carbons each, i.e., 36 carbons per non-polar domain in the lamellar structure. Consequently, using Tanford’s equation [41], the maximum thickness of the DODAB non-polar domain in the gel state can be found to be 48 Å. The experimental value of the DODAB, d_sp_, was considerably smaller in the gel phase, which suggested the existence of lamellar structures, with chains adopting a tilted and/or interdigitated configuration.

DODAB arranges itself into a swollen lamellar phase (d_sp_ = 126 Å) when heated above its melting temperature, resulting in a bilayer thickness of 30 Å and a surfactant molecule area of 68 Å^2^. This is in line with the expected decrease from a “solid-like” to “fluid-like” hydrocarbon state, assuming that all chains are in an all-trans conformation, and are perpendicular to the interface in the L_β_ phase, and are in a liquid hydrocarbon state and perpendicular to the interface in the L_α_ phase [45,46].

#### 2.2.3. Salt Effects

The presence of electrolytes in the solution will alter the electrostatic interactions between surfactant headgroups, which in turn may affect the thermal transitions. Here, we investigated the effects of sodium bromide (NaBr) and sodium chloride (NaCl) on the bilayers of DODAB and DODAC, respectively. Figure 7 presents the thermal analysis and SAXS characterisation of the bilayers of DODAB and DODAC in the presence of salt, and the T_m_, enthalpy of change and interlamellar spacing values are tabulated in Table 4 and Table 5. The T_m_ remained unchanged for both surfactants, but a slight decrease of the enthalpy of phase transition on melting was detected. The d_sp_ of DODAB’s L_β_ phase was unchanged, while a more pronounced effect in the L_α_ phase was detected. Firstly, the recorded d_sp_ of DODAB in the L_α_ phase was 126 Å, which is in good agreement with the calculated d_max_ using Equation (4), 123 Å. Secondly, the addition of salts decreased the repulsion between headgroups, which resulted in the maintenance of a highly swollen L_α_ phase. Likewise, a similar trend for DODAC bilayers is found. The calculated d_max_ of DODAC in the L_α_ phase is 114 Å, and a d_sp_ of 84 Å was recorded. NaCl screens the headgroup charges, which resulted in a closer packing of the surfactant molecules, thus a further swelling to 95 Å was recorded. The same behaviour was also observed for the L_β_ phase.

### 2.3. Additive Effects

#### 2.3.1. Effects on Bilayer Stability and Packing

As described in the Introduction, surfactant formulations are complex mixtures, and contain additives with a wide range of polarity. In order to provide a general understanding of additive effects in double-chain surfactants systems, as well as to give a basis for formulation work, we have investigated how additives impact a number of aspects of DODAC-water lamellar phases. We studied 15 different low-molecular-weight additives with widely different structure and polarity, including a range from highly water-soluble to water-insoluble substances; the chemical structures are given in Figure 8 and the logarithm of the octanol-water partition coefficient is tabulated in Table 6. Using DSC, we investigated the phase transition temperature, as well as the associated enthalpy change. The L_α_ and L_β_ phases were investigated by SAXS and WAXS, and the packing was characterised by calculating the bilayer thickness and the average head-group area as described above. With some additives, the L_α_ and L_β_ phases were retained, while in others, an alternative phase was formed on the melting of L_β_. Phases formed included the hexagonal phase (Hex), cubic phase (Cub) and sponge phase (L3); our study was focused on the lamellar phases, and closer investigations of other phases were not performed. Furthermore, we investigated the reversibility of the additives’ incorporation by dilution studies.

Based on previous work on single-chain ionic surfactant systems, a number of different scenarios can be predicted depending on additive chemical structure [9]:polar highly water-soluble cosolutes can reduce the hydrophobic interactions, which drive surfactant self-assembly;oppositely charged amphiphilic ions may strengthen association by weakening opposing electrostatic interactions;electrolytes may screen electrostatic repulsions and strengthen association (electrolyte addition is dealt with in Section 2.2.3);amphiphilic nonionic cosolutes may strengthen association by decreasing the charge density of the aggregates.

We can distinguish between two cases of additive effects: one case where even up to quite high additive concentrations, it is still possible to observe a L_β_ to L_α_ transition; and another case where an alternative phase is found at an increasing temperature. The chain packing effects of additives in the former case is the one where it is most straightforward to discuss.

In cases where we observe a L_β_ to L_α_ transition over wide ranges of additive concentration, we note that the geometrical measures characterising surfactant packing mainly change for the L_α_ phase, while only minor effects of additives are noted for the L_β_ phase. As expected, a fluid-like phase would be more sensitive to additives than a solid-like phase. On this ground, we can also attribute changes in the transition temperature to changes in the stability of the L_α_ phase.

Figure 9 shows the effect of the 15 additives on the main thermal transition under heating. As seen in Figure 9a, urea and two simple urea derivatives have no significant effect on the transition temperature, even up to high concentrations. Furthermore, they do not affect the enthalpy of the thermal transition (Appendix A). These compounds also do not significantly affect the X-ray diffraction patterns; thus, the bilayer thickness and the area per surfactant molecule in the bilayer are the same as in the absence of additive. Thus, replacing water with urea and these two urea derivatives seems to have no appreciable effect on interactions between the surfactant molecules. We note that for more weakly associating systems, which are on the balance of association, urea is well-known to have significant effects, such as the micelle formation of relatively polar surfactants, or the self-assembly of DNA (into the double helix) or cellulose [12,48,49,50,51,52,53,54,55,56,57].

For the other additives studied, the situation is different – several additives caused major changes in the main transition temperature (Figure 9 and Appendix A) and packing (Figure 10). Alcohols and fatty acids are known to have large effects on the self-assembly of ionic surfactants, with quantitative differences depending on the polarity of the additive. A highly polar compound is located mainly in the aqueous region due to high solubility, and this will change the solvent character, leading to weakened hydrophobic interactions – therefore, in this case, the CMC will increase. For an alcohol or fatty acid with a longer alkyl chain, a completely different pattern is observed; these compounds have a lower aqueous solubility, and will be located in the non-polar parts of the aggregates. Thus, the CMC is markedly decreased, and furthermore, major changes in surfactant packing, leading to changes in the aggregates, are observed. Therefore, these additives may induce changes from spherical micelles to thread-like ones or hexagonal or lamellar liquid crystalline phases at higher additive concentrations.

In Figure 10, we present the results on the L_β_ to L_α_ transition for four different fatty acids, two aromatic alcohols and five fatty alcohols. The data presented include the lamellar spacing obtained from the SAXS patterns, from which the bilayer thickness and the area per polar head-group were calculated as described above (refer to Equations (1)–(3)). As can be seen, the effects are completely different for different additives, but following the outlined expectations. Medium-chain compounds may induce a major lowering of the transition temperature, and the same change is observed for the enthalpy. We ascribe this to the stabilisation of the L_α_ phase, due to the lowering of the electrostatic penalty in the self-assembly process; this will be discussed further next. Long-chain alcohols have a strikingly opposite effect in increasing the transition temperature; long-chain alcohols act as a second surfactant, and mixed aggregates are formed.

Another additive that was studied is sodium butyrate (Figure 9b) – this has a very different character, as it contains a weakly amphiphilic ion of opposite charge to the surfactant; furthermore, it can increase the ion concentration in the aqueous regions. The butyrate ion will be included in the surfactant head-group region, and the ions will screen the electrostatic repulsions. Indeed, sodium butyrate shows very different effects compared to the other additives. From the SAXS data, a major change in packing can be inferred at the same time as small effects are noted for the transition temperature and the enthalpy (Appendix A). As we have discussed previously, the very thin surfactant layers suggest a major interpenetration and tilting in the bilayer. Regarding the transition temperature and enthalpy, we assign the small changes to a compensation of electrostatic screening in the L_β_ phase and the lowering of aggregate charge density in the L_α_ phase.

#### 2.3.2. Transitions to Nonlamellar Phases

As described above, DODAC forms lamellar phases over a wide range of concentrations; at low temperature, there is a L_β_ phase with solid-like surfactant alkyl chains, and a L_α_ phase with liquid-like chains above the T_m_. Adding cosolutes of different kinds may shift the transition temperature to a smaller or larger extent, but with most additives studied, the L_α_ and L_β_ phases are still present, even at quite high additive concentrations. This work is focused on the transition between the lamellar phases of the double-chain cationic surfactant. In a few cases, there is a change in phase structure, which we will now briefly consider. Previously we have discussed a transition to a liquid bicontinuous solution (L3) for sodium butyrate [21].

Cubic liquid crystals were identified in systems with hexanoic acid and benzyl alcohol, but the detailed structure was outside the scope of this study. For butanol, we found a similar behaviour as in the case of sodium butyrate. The X-ray scattering patterns shown in Figure 11 thus demonstrate the presence of strongly smeared reflections above T_m_. With longer fatty alcohols, other structures were found, including cubic liquid crystals for hexanol and octanol and reversed hexagonal liquid crystals for decanol and dodecanol (Appendix A).

The effect of butanol on the lamellar phase has a clear correlation with its role in the formation of bicontinuous microemulsions for single-chain surfactants [58,59]. These microemulsions are typically composed of an ionic surfactant, alkane, water and butanol. The microstructure can be of the O/W, W/O or bicontinuous type, depending on conditions like salinity, but is over wide ranges of the bicontinuous type, as we find here and as illustrated in Figure 11d. At a temperature above the thermal transition (50 °C), the first and most intense reflection is at 0.1015 Å^−1^, followed by a shoulder at around 0.2002 Å^−1^ (Figure 11d). Though not completely clear at this stage, the scattering pattern suggests a lamellar phase with a d_sp_ of 62 Å. The presence of a high amount of butanol in the system may force the surfactant to adopt a reversed structure. Currently, we cannot rule out entirely the possible existence of reversed hexagonal domains, which would contribute to the scattering pattern. Therefore, further studies of the effect of butanol on the liquid crystalline phase of DODAC are recommended. Analogous scattering patterns are found for microemulsions [60].

In microemulsion systems, there is often a delicate balance between a lamellar liquid crystalline phase and bicontinuous microemulsions. The two structures are both characterised by a CPP value of around 1, but differ in the flexibility of the surfactant film. An important role of a relatively short-chain alcohol is to lower the rigidity of the surfactant film. To underline even more the connection to the present findings, we note that short-chain alcohols can induce similar changes for lamellar lipids, such as a transition from lamellar liquid crystalline phase to bicontinuous microemulsions for lecithin [61].

Surfactant packing is normally discussed in terms of the spontaneous curvature of the surfactant film or the critical packing parameter (CPP), as Figure 12 schematically illustrates. As we reduce the electrostatic interactions in ionic surfactant systems, the CPP increases, and for lamellar phases, we expect then transitions to so-called reversed structures (with a CPP above 1 and a negative spontaneous curvature) like the L3 phase, reversed cubic or reversed hexagonal liquid crystals. While corresponding effects have been investigated in detail for single-chain ionic surfactants, including phase diagrams and microstructure, the information is sparse in literature for double-chain ionic surfactants. However, one example is given in phase diagram studies of a double-chain anionic surfactant, Aerosol OT. Fontell noted that on the addition of salt, there is a transition to a L3 phase [62]. In our case, we noted an analogous transition for sodium butyrate, and discussed it in a previous publication [21].

A few interesting observations are made in the presence of long-chained alcohols. These additives act as cosurfactants, which contribute to the stability of lamellar phases. Figure 13 shows the packing structure of DODAC in the presence of octanol and dodecanol at temperatures below and above the T_m_ using SAXS. Below the T_m_, dodecanol was incorporated in the DODAC bilayers, and a bilayer thickness of 38 Å in the gel phase was determined (Figure 13a,d). We remind the reader that the d_bi_ of pure DODAC in water was determined to be 24 Å. Therefore, dodecanol had significantly increased the stability of the L_β_ phase, and reduced the tilting or interdigitation of the alkyl chains, consequently resulting in a higher T_m_ (Figure 9). Upon heating, the melting of the L_β_ phase developed a reversed hexagonal phase (Figure 13c,f). The increase in temperature makes the long dodecanol hydrocarbon chain even less polar, and forces the surfactant system to adopt a reversed hexagonal structure, where polar channels are surrounded by non-polar chains in the fluid-like state, schematically depicted in Figure 13h. Furthermore, the X-ray reflections allowed to determine the polar channels’ radius, and no variation was identified for those systems that developed a reversed hexagonal phase (Table 7). The melting of the L_β_ phase with octanol has developed a cubic phase coexisting with a reversed hexagonal (Figure 13b,e). The evolution to a bicontinuous cubic phase upon heating is somewhat expected due to the alkyl chain length of octanol, which eventually resulted in single hexagonal phases in the presence of long-tailed alcohols such as dodecanol. For single-chain ionic surfactants, an analogous effect of long-chain alcohols is well-established [15].

#### 2.3.3. Dilution Experiments

Of practical significance for some applications but also of interest with respect to the molecular interactions is the resistance of the incorporation of the additive in the surfactant structure. To this end, we monitored structural changes on large dilutions with water. This was dealt with for a few additives in a previous publication [22], and here only some brief comments are made. The ternary surfactant-additive-water mixtures were diluted with water on a 1:10 sample/water ratio. These diluted mixtures exhibit a rather thick water layer, and therefore they were characterised using the higher brilliance synchrotron small-angle X-ray scattering beamline, as the scattering is rather weak. An important observation is that if we dilute a sample with an additive with a considerably lowered transition temperature, the SAXS results show that we still have a lamellar structure (Figure 14). Furthermore, we find that the diluted sample shows a thermal transition corresponding to the surfactant-water system without additive. We can thus deduce that the additive has been removed from the surfactant bilayers.

Above we showed that the addition of 12 wt% octanol forced the DODAC bilayer to change to a reversed bicontinuous phase. However, here we demonstrate that the same additive can be removed from the surfactant’s bilayers with water, and the original single lamellar phase is re-established. On extensive dilution of the DODAC-octanol suspension, synchrotron SAXS revealed a highly swollen L_β_ phase (d_sp_ of 338 Å) and L_α_ phase (d_sp_ of 330 Å) at a temperature below and above the T_m_, respectively (Figure 14b). This piece of evidence supported the hypothesis that additives can be selectively chosen to alter the phase behaviour, and subsequently removed on dilution with water to restore the initial phase without additive, as schematically depicted in Figure 14c.

This observation opens the possibility of designing formulations with spontaneous transition from L_α_ to L_β_ on dilution. For example, double-chain cationic surfactants are efficient as conditioners and softeners (hair, textiles) in the L_β_ form. In commercial formulations, the surfactant occurs as a kinetically stable dispersion of the L_β_ phase in water. On the other hand, it is easy to create thermodynamically stable formulations containing the L_α_ phase over wide composition ranges. It would therefore be of interest to have thermodynamically stable systems of L_α_, which on dilution spontaneously transform into the L_β_ phase. Our dilution experiments clearly indicate this possibility. Thus, on dilution, we observe the reappearance of the characteristic WAXS diffraction pattern of the L_β_ phase, in addition to the lamellar diffraction patterns. In a previous publication, we have in some detail analysed this problem, and also presented deposition studies monitoring deposition on a substrate by in-situ null ellipsometry [22].

## 3. Conclusions

There is a strong tendency for double-chain surfactants and lipids to self-assemble in aqueous systems, and this can start at low concentrations. Self-assembly of the surfactants can lead to bilayer structures, particularly of the lamellar type, over wide ranges of composition. Lamellar phases can be of two principal types, depending on the state of the surfactant alkyl chains. In the lamellar gel phase, L_β_, the chains are in a “frozen” all-trans conformation, while in the lamellar liquid crystalline phase, L_α_, they are in a “melted” liquid-like state. For a cationic surfactant dioctadecyldimethylammonium chloride (DODAC), and to some extent for the corresponding bromide (DODAB), the two phases and the phase transition were characterised by differential scanning calorimetry and X-ray diffraction – SAXS and WAXS – for a wide range of compositions of surfactant-water mixtures. In addition, the effect on phase transitions and phase structures of a large number of additives was investigated.

In contrast to a simple picture of surfactant packing in the bilayers, the bilayer spacing in L_β_ is found to be smaller than that in L_α_, which excludes a structure with surfactant molecules being oriented perpendicularly to the bilayer; instead, a structure with tilted or interdigitated alkyl chains is inferred. From the SAXS diffraction patterns, it could be concluded that there is large water swelling, and that no significant changes in the bilayer structures occur when the water content is varied over wide ranges.

A number of additives of very different natures were investigated, with respect to their effect on the stability and structure of the lamellar phases. Certain additives, such as urea, have no significant effect on either the stability or the surfactant packing. Alcohols and fatty acids have varying effects that can be rationalised in terms of polarity; for example, an alcohol with a long alkyl chain will act as a second surfactant, and change the packing and interactions notably. Medium-chain fatty acids and alcohols markedly decrease the transition temperature between gel and liquid crystalline phases, while long-chain alcohols, acting as a non-ionic surfactant, do not have this effect.

On extensive dilutions with water, the additives are dissociated from the bilayers, leading to a restoration of the phase transition observed for surfactant alone. This opens the possibility for applications where the L_α_ phase, which is more stable and easier to formulate, spontaneously converts to L_β_ upon dilution with water. In this way, the L_β_ phase, which is rendering conditioning effects in hair applications, can be formed in-situ on a substrate.

While in most cases the two lamellar phases persist up to high additive concentrations, in a number of cases, transitions to other phases occur. Cubic liquid crystals were identified with hexanoic acid and benzyl alcohol. For butanol, we found a bicontinuous liquid phase, while with longer fatty alcohols, other structures were found, cubic liquid crystals for hexanol and octanol and reversed hexagonal liquid crystals for decanol and dodecanol. The phase changes can be predicted using conventional models of balance between hydrophobic and hydrophilic interactions.

## 4. Materials and Methods

Dioctadecyldimethylammonium chloride (DODAC, 96.7% purity) was supplied by Evonik Corporation, USA. Dioctadecyldimethylammonium bromide (DODAB, 98.0% purity), urea, 1-methyl urea, 1,3-dimethyl urea, and sodium butyrate were purchased from Tokyo Chemical Industry Co., Ltd, Tokyo, Japan. Acetic acid, propionic acid, butyric acid, hexanoic acid, 1-butanol, 1-hexanol, 1-octanol, 1-decanol and 1-dodecanol were purchased from Sigma-Aldrich, Singapore, Singapore. Benzyl alcohol was supplied by Ineos Chlorotoluenes, Tessenderlo, Belgium, and phenoxyethanol supplied by Clariant Produkte GmbH, Frankfurt, Germany. Ultrapure water of 18 S/m conductivity was used to prepare the samples. Binary DODAB/C–water and ternary DODAB/C–water–additive mixtures were prepared, as described previously [22]. Diluted samples were prepared by weighing a fraction of the original sample and diluting in a 1:10 sample/water ratio, in a screw-cap glass vial. All the samples were equilibrated at room temperature for at least seven days before characterisation.

### 4.1. Polarised Optical Microscopy

The phase’s mosaic texture observation was attained under polarised light, resulting in a fast and straightforward method to determine the phase structure. Polarised optical microscopy (POM) is one of the few inexpensive methods often used to determine whether birefringence is present in the prepared samples. To perform POM, a drop of the mixture was deposited on a glass slide and covered with a cover glass. All micrographs were obtained using an Olympus BX53 (Olympus Corporation, Singapore, Singapore) optical microscope with a polariser filter, coupled with an Infinity 1 camera (Lumenera, Singapore, Singapore). The micrographs were analysed using Infinity Analyse software (Lumenera).

### 4.2. Differential Scanning Calorimetry

The main thermal transition upon melting (T_m_) and cooling (T_c_) were determined by using a Discovery DSC differential scanning calorimeter (TA Instruments, Singapore, Singapore) at a rate of 2 °C/min from 10 to 60 °C and back to 10 °C. The chamber was kept under a nitrogen environment. The software Trios (TA Instruments) was used to calculate the enthalpy associated with the phase transitions (ΔH_m_ and ΔH_c_). The enthalpy associated with a thermal transformation was determined by integrating the endothermic or exothermic peak areas.

### 4.3. Small- and Wide-Angle X-ray Scattering

SAXS and WAXS characterisation was carried out using two X-ray source instruments. A simultaneous SAXS/WAXS laboratory instrument, Nano-inXider (Xenocs, Grenoble, France), equipped with a micro-focus source generating X-rays of a wavelength λ = 1.542 Å (Genix3D), operating at 50 kV and 0.6 mA, was used to characterise the original mixtures. The small and wide-angle X-ray scattering beamline at the Australian Synchrotron operating with incident X-rays beam of a wavelength λ = 1.512 Å (8.2 keV beam) with a Pilatus 2M detector located at 7000 mm was used for SAXS. An incident X-rays beam of a wavelength λ = 0.827 Å (15 keV beam) with a Pilatus 2M detector located at 320 mm was used for WAXS. Fluid samples were loaded into thin-walled borosilicate capillaries (1.5 mm outer diameter, 0.01 mm wall thickness, Hampton Research), and mounted to a multi-capillary block connected to a water bath for thermal behaviour studies. The system was calibrated using a AgBeh standard. Each sample was exposed to the beam for 600 s using the Nano-inXider, and 1 s at the synchrotron. The resulting 2D scattering patterns were integrated to 1D background-corrected patterns using XSACT software (Xenocs) and ScatterBrain software (Australian Synchrotron, Clayton, Australia).

## Figures and Tables

**Figure 1 molecules-26-03946-f001:**
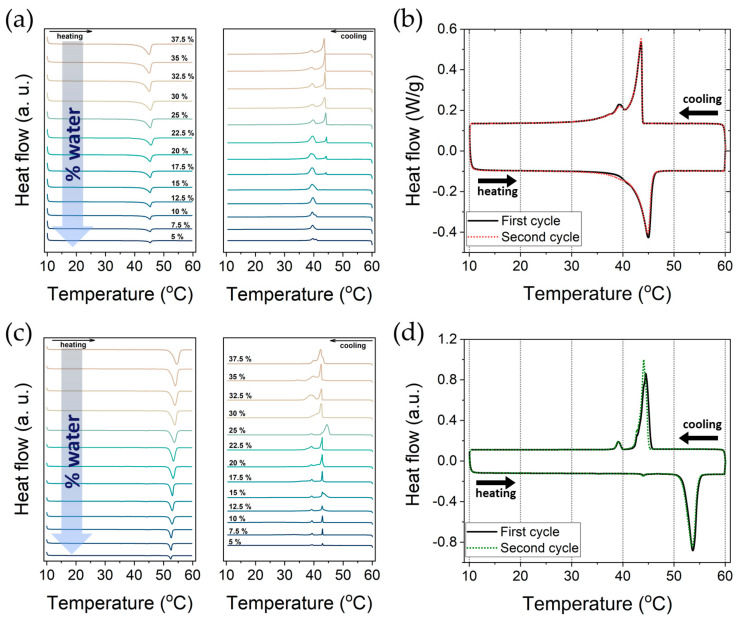
Thermal analysis under heating and cooling of two double-chain cationic surfactants. Thermograms from 5 to 37.5 wt% of (**a**) DODAC and (**c**) DODAB in water. Two-cycle thermograms under heating and cooling of (**b**) 35 wt% DODAC and (**d**) 25 wt% DODAB in water. The full line corresponds to the first cycle and the dotted line to the second cycle. The measurements were conducted from 10 to 60 °C, and back to 10 °C at 2 °C/min. The endothermic event corresponds to the downwards deflection.

**Figure 2 molecules-26-03946-f002:**
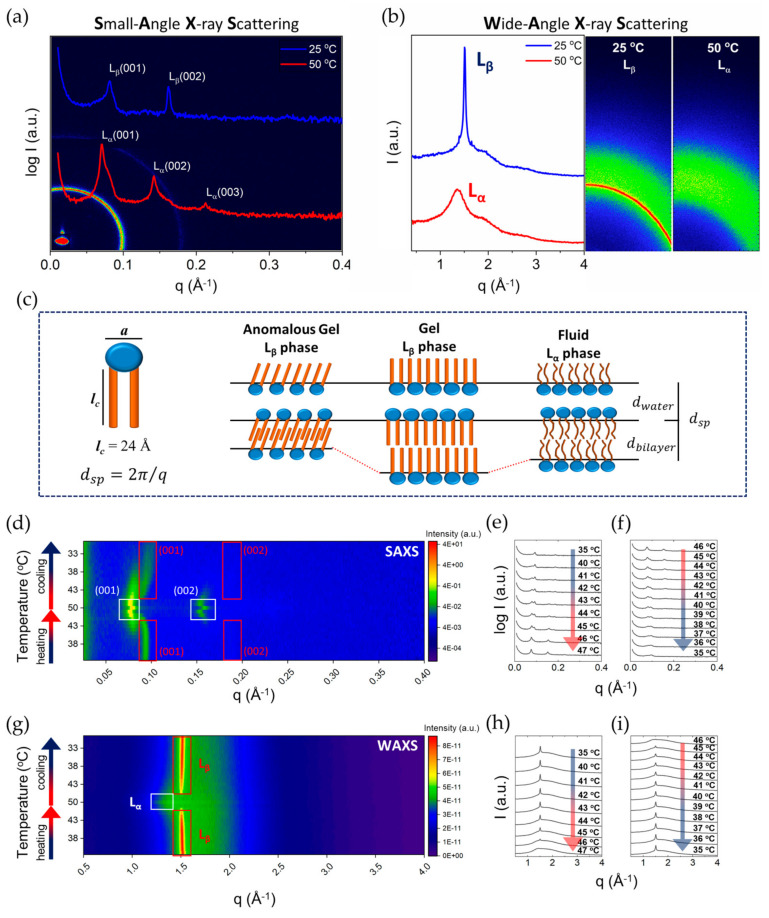
Small- and wide-angle X-ray scattering characterisation of 35 wt% DODAC in water. (**a**) SAXS and (**b**) WAXS at a temperature below (25 °C) and above (50 °C) the T_m_. (**c**) Schematic representation of various lamellar phases. a—area per surfactant molecule, l_c_—hydrocarbon chain length, d_sp_—interlamellar spacing. (**d**) 2D image and corresponding (**e**,**f**) 1D SAXS patterns and (**g**) 2D image and corresponding (**h**,**i**) 1D WAXS patterns during a heating and cooling ramp from 35 to 50 °C, and back to 35 °C. L_β_ corresponds to the gel phase and L_α_ to the liquid crystalline phase.

**Figure 3 molecules-26-03946-f003:**
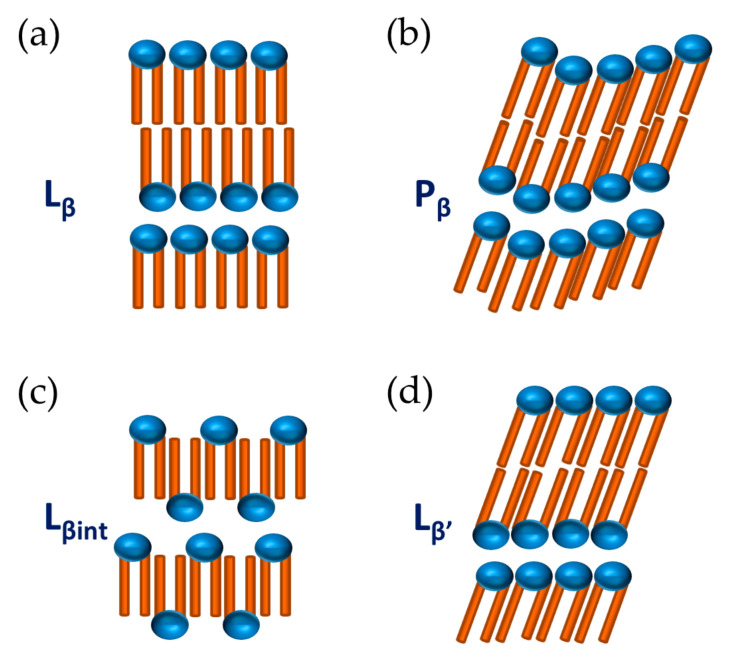
Various structures of bilayers in the gel phase. (**a**) Gel phase, L_β_; (**b**) Rippled gel phase, P_β_; (**c**) Interdigitated gel phase, L_β__int_; (**d**) Tilted gel phase, L_β_’.

**Figure 4 molecules-26-03946-f004:**
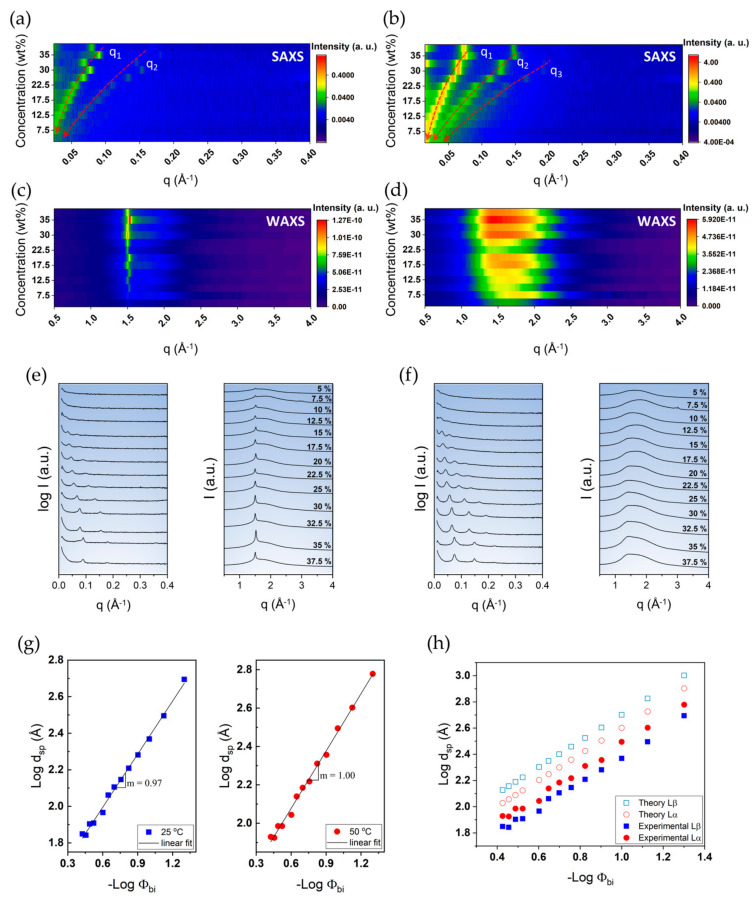
Small- and wide-angle X-ray scattering characterisation of DODAC in water from 5–37.5 wt% at a temperature below (25 °C) and above (50 °C) the T_m_. (**a**,**c**) 2D images and corresponding (**e**) 1D SAXS and WAXS patterns at 25 °C. (**b**,**d**) 2D images and corresponding (**f**) 1D SAXS and WAXS patterns at 50 °C. (**g**) Double logarithmic plot of interlamellar spacing (d_sp_, Å) versus inverse bilayer volume fraction (Φ_bi_) for DODAC in water at 25 °C and 50 °C. (**h**) Double logarithmic plot of interlamellar spacing (d_sp_, Å) versus inverse bilayer volume fraction (Φ_bi_) for DODAC in water. Empty symbols correspond to the theoretical maximum interlamellar spacing (d_max_, Å). Full-colour symbols correspond to the experimental results calculated from SAXS and WAXS analysis of the L_β_ gel phase (25 °C, square) and L_α_ liquid crystalline phase (50 °C, circle).

**Figure 5 molecules-26-03946-f005:**
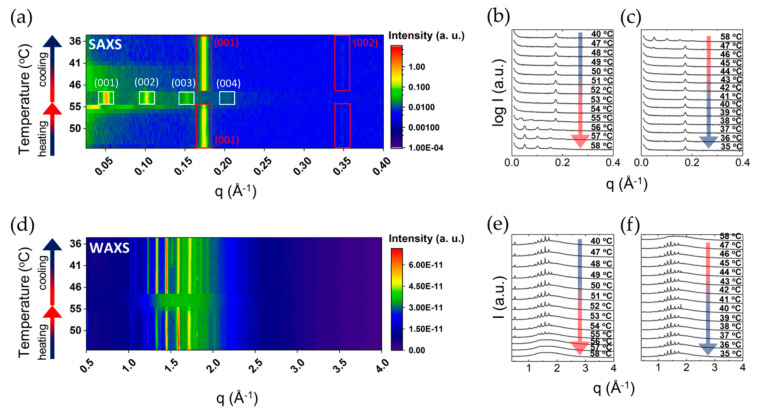
Small- and wide-angle X-ray scattering characterisation of 25 wt% DODAB in water. (**a**) 2D image and corresponding (**b**,**c**) 1D SAXS patterns and (**d**) 2D image and corresponding (**e**,**f**) 1D WAXS patterns during a heating and cooling ramp from 40 to 58 °C and back to 35 °C.

**Figure 6 molecules-26-03946-f006:**
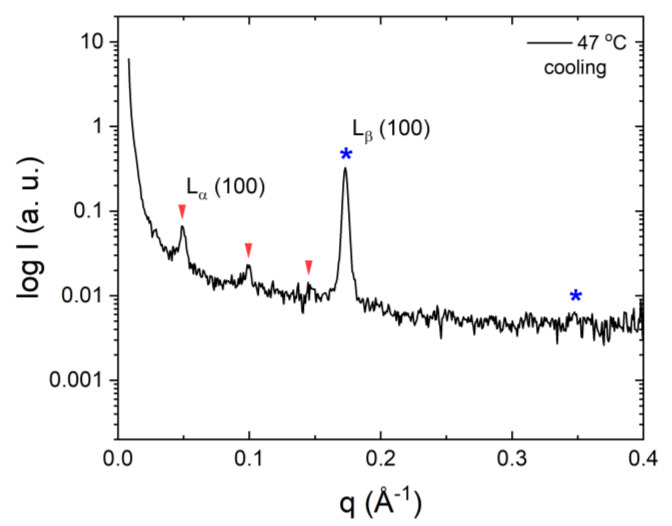
SAXS pattern of 25 wt% of DODAB in water at 47 °C under cooling showing the coexistence of two lamellar phases. The symbols represent lamellar reflections; arrows the liquid crystalline phase and stars the gel phase.

**Figure 7 molecules-26-03946-f007:**
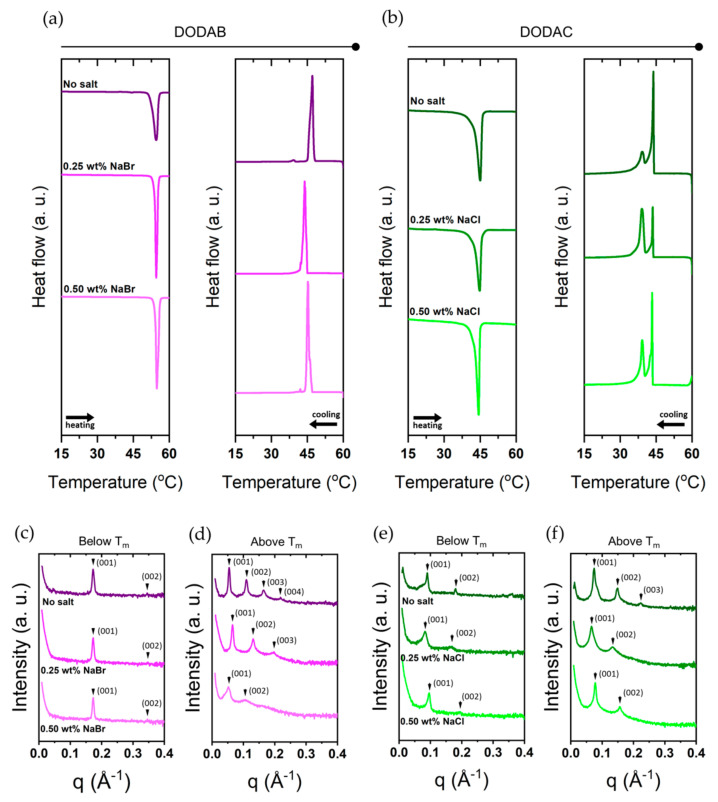
Thermal and packing characterisation of 35 wt% DODAB and 35 wt% DODAC in the presence of electrolytes. (**a**,**c**,**d**) DODAB in the presence of 0.25 wt% (0.08 M) and 0.50 wt% (0.16 M) of NaBr; (**b**,**e**,**f**) DODAC in the presence of 0.25 wt% (0.09 M) and 0.50 wt% (0.19 M) of NaCl. SAXS patterns were collected below (25 °C) and above (65 °C) the T_m_.

**Figure 8 molecules-26-03946-f008:**
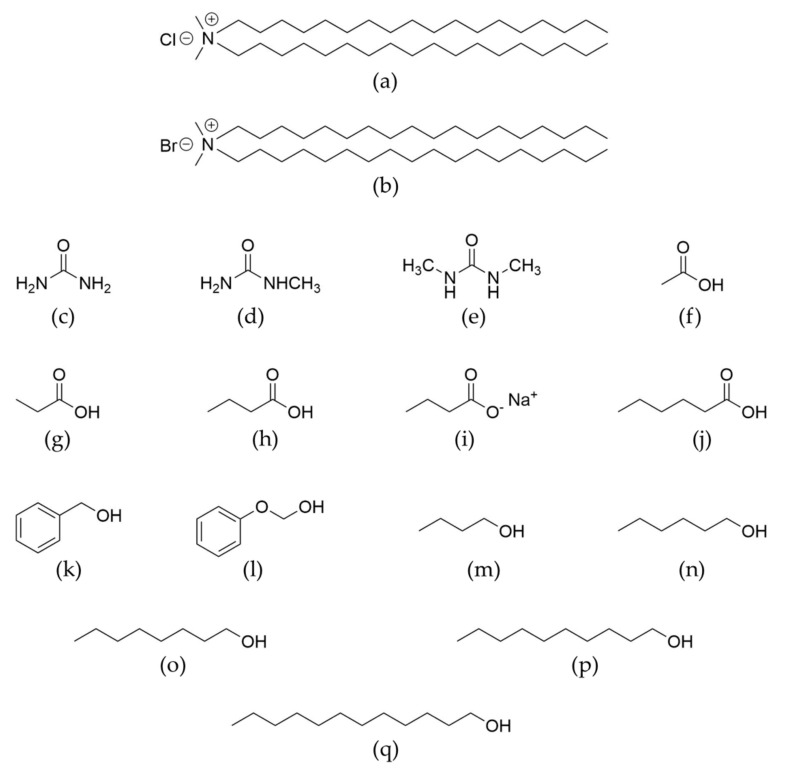
Molecular structure of: (**a**) dioctadecyldimethylammonium chloride—DODAC; (**b**) dioctadecyldimethylammonium bromide—DODAB; (**c**) urea—U; (**d**) methyl urea—MU; (**e**) dimethyl urea—DMU; (**f**) acetic acid—AA; (**g**) propionic acid—PA; (**h**) butyric acid—BA; (**i**) sodium butyrate—SB; (**j**) hexanoic acid—HA; (**k**) benzyl alcohol—BenOH; (**l**) phenoxyethanol—PhEtOH; (**m**) butanol—ButOH; (**n**) hexanol—HexOH; (**o**) octanol—OctOH; (**p**) decanol—DecOH; and (**q**) dodecanol—DodecOH.

**Figure 9 molecules-26-03946-f009:**
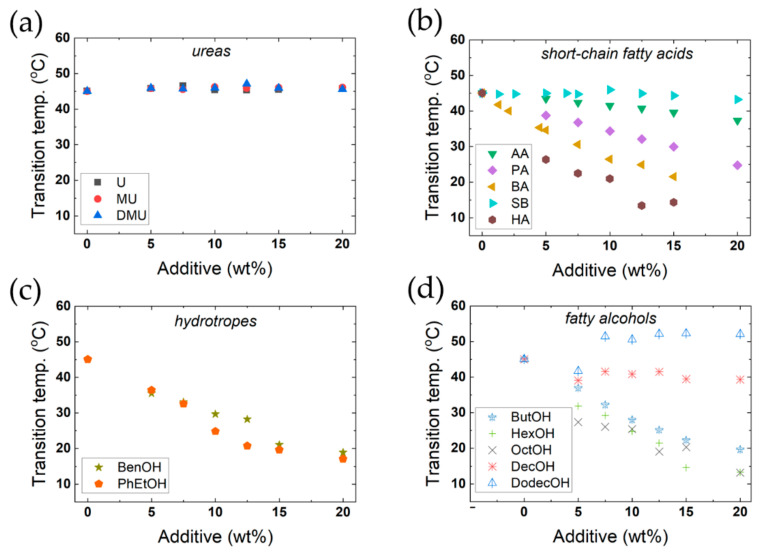
Effect of added additives on the main phase transition temperature of DODAC in water. (**a**) Urea and two urea derivatives. U-urea, MU-methyl urea, DMU-dimethyl urea. (**b**) Short-chain fatty acids and sodium butyrate. AA-acetic acid, PA- propionic acid, BA- butyric acid, SB-sodium butyrate, HA-hexanoic acid. (**c**) Hydrotrope molecules. BenOH-benzyl alcohol, PhEtOH-phenoxyethanol. (**d**) Fatty alcohols. ButOH-butanol, HexOH- hexanol, OctOH- octanol, DecOH-decanol, DodecOH-dodecanol.

**Figure 10 molecules-26-03946-f010:**
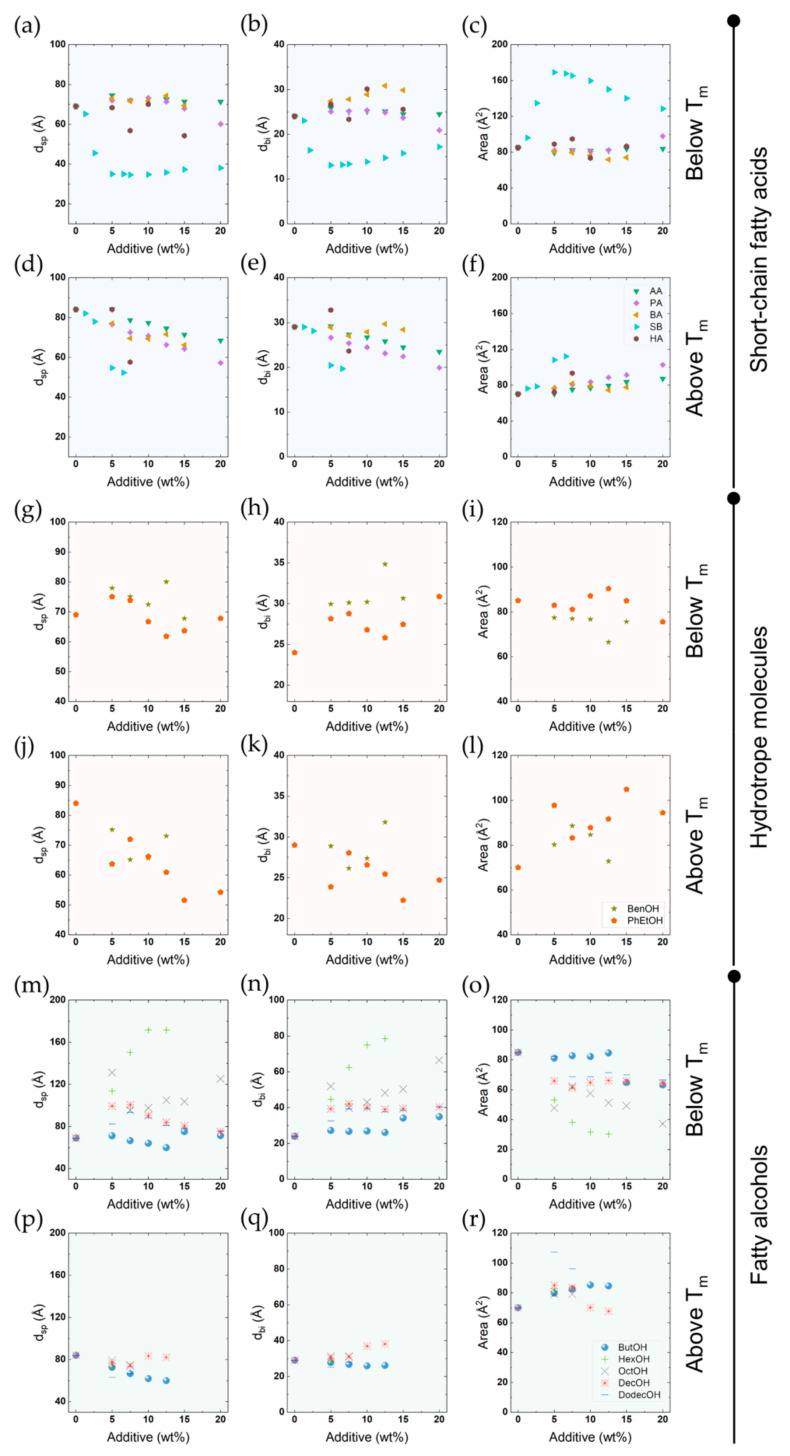
Effects of additives in the interlamellar spacing (d_sp_), bilayer thickness (d_bi_) and area per surfactant molecule of the DODAC lamellar structure. (**a**–**f**) Short-chain fatty acids: AA-acetic acid, PA-propionic acid, BA-butyric acid, SB-sodium butyrate, HA-hexanoic acid. (**g**–**l**) Hydrotrope molecules: BenOH-benzyl alcohol, PhEtOH-phenoxyethanol. (**m**–r) Fatty alcohols: ButOH-butanol, HexOH-hexanol, OctOH-octanol, DecOH-decanol, DodecOH-dodecanol. The top panels (**a**–**c**,**g**–**i**,**m**–**o**) correspond to the gel phase, and the bottom panels (**d**–**f**,**j**–**l**,**p**–**r**) correspond to the liquid crystalline phase.

**Figure 11 molecules-26-03946-f011:**
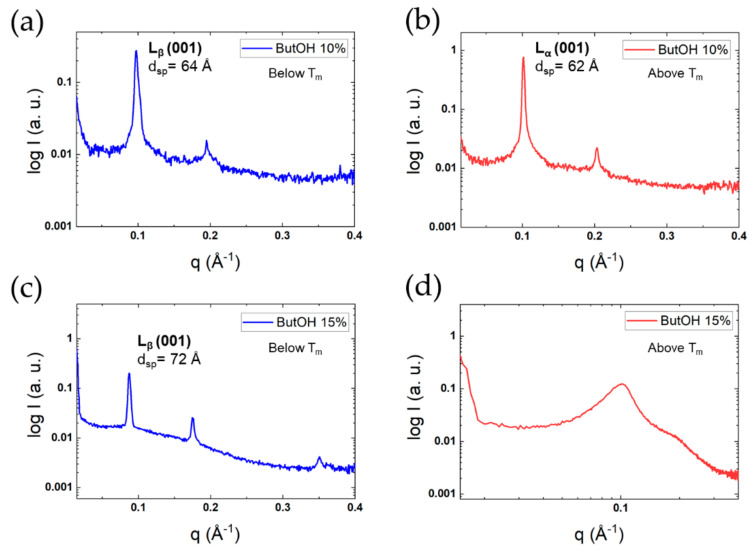
SAXS patterns for DODAC in the presence of butanol. Top panels: 10.0 wt% of the additive: (**a**) below the T_m_, and (**b**) above the T_m_. Bottom panels: 15.0 wt% of the additive: (**c**) below the T_m_ (10 °C) and (**d**) above the T_m_.

**Figure 12 molecules-26-03946-f012:**
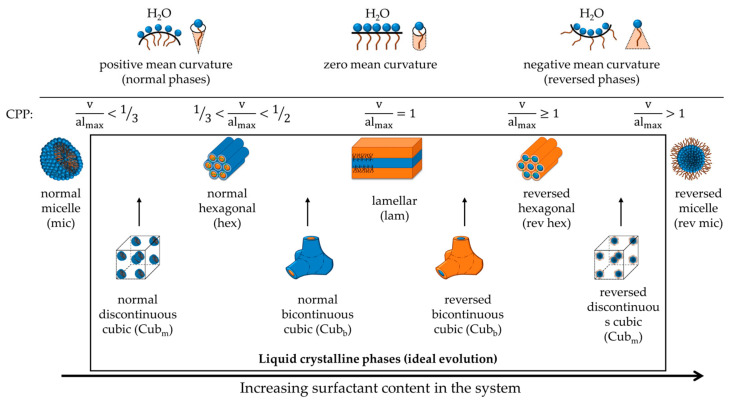
Representation of the surfactant self-assembly evolution at different values of the critical packing parameter (CPP) and the geometrical constraints. The CPP relates the head group area (a), the extended length (l_max_) and the volume (v) of the hydrophobic part of a surfactant molecule. The box highlights the various liquid crystalline phases.

**Figure 13 molecules-26-03946-f013:**
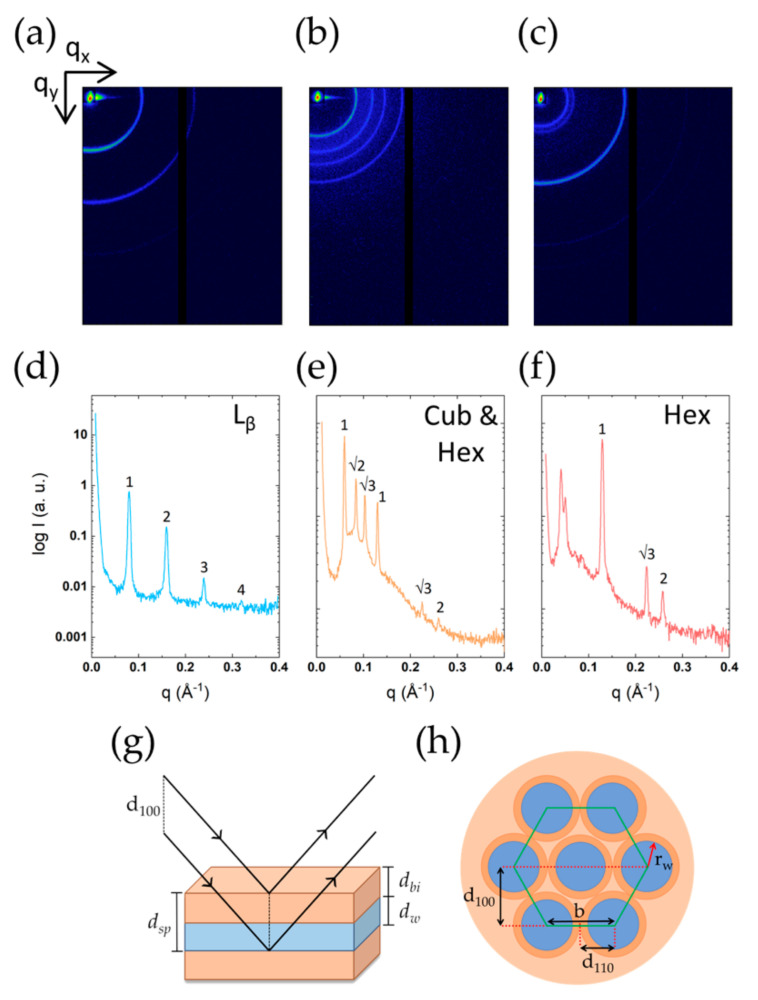
The various geometries exhibited by DODAC in the presence of different alcohols. Top panel (**a**–**c**): 2D images, middle panel (**d**–**f**): integrated 1D scattering patterns, and bottom panel (**g**,**h**): schematic representation of a lamellar and a hexagonal structure and the information obtained from their scattering pattern. (**a**,**d**) 35 wt% DODAC-15 wt% dodecanol at 25 °C—L_β_ phase; (**b**,**e**) 35 wt% DODAC-12.5 wt% octanol at 50 °C—Cub and (reversed) Hex phases; (**c**,**f**) 35 wt% DODAC-20 wt% dodecanol at 50 °C—(reversed) Hex phase.

**Figure 14 molecules-26-03946-f014:**
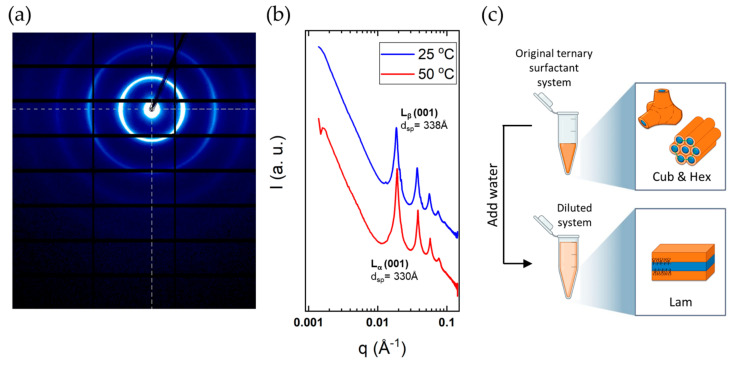
SAXS analysis of diluted (10 times) 35 wt% DODAC-12.5 wt% Octanol in water. (**a**) 2D scattering pattern at 50 °C and (**b**) corresponding 1D integrated pattern at 25 °C and 50 °C. (**c**) Scheme of the phase evolution on dilution with water.

**Table 1 molecules-26-03946-t001:** Normalised differential scanning calorimetry parameters of the DODAB and DODAC in water from 5 to 37.5 wt% of surfactant.

	DODAB		DODAC	
Surfactant (wt%)	Heating Cycle		Heating Cycle	
	T_m_ (^°^C)	ΔH_m_ (kJ mol^−1^)	T_m_ (°C)	ΔH_m_ (kJ mol^−1^)
5.0	52.5	63.6	45.4	40.2
7.5	52.6	94.7	45.3	42.5
10.0	52.6	96.7	45.3	39.2
12.5	53.0	97.2	45.3	41.4
15.0	53.0	98.0	45.3	42.9
17.5	53.0	95.4	45.3	46.0
20.0	53.3	109.6	45.3	46.0
22.5	53.4	113.0	45.6	46.7
25.0	53.6	108.2	45.4	45.9
30.0	53.8	99.6	45.4	47.5
32.5	53.9	102.3	45.1	44.3
35.0	54.0	109.3	45.1	45.8
37.5	54.5	102.7	45.0	44.4

**Table 2 molecules-26-03946-t002:** Calculated values of the bilayer volume fraction (Φ_bi_), the interlamellar spacing (d_sp_), bilayer thickness (d_bi_), polar layer thickness (d_w_) and area per surfactant molecule (a) of DODAC bilayers in water at temperatures below (25 °C) and above (50 °C) the main phase transition temperature.

		25 °C				50 °C				
DODAC (wt%)	Φ_bi_	d_sp_ (Å)	d_bi_ (Å)	d_w_ (Å)	a (Å^2^)	d_sp_ (Å)	d_bi_ (Å)	d_w_ (Å)	a (Å^2^)
5.0	0.052	495	26	469	80	600	31	569	66	
7.5	0.079	313	25	288	83	400	32	369	65
10.0	0.107	234	25	209	82	313	33	279	62
12.5	0.128	191	24	167	84	227	29	198	71
15.0	0.154	162	25	137	82	205	32	173	65
17.5	0.179	140	25	115	82	165	30	135	69
20.0	0.204	127	26	101	79	153	31	122	66
22.5	0.229	115	26	89	78	138	32	106	65
25.0	0.253	92	23	69	88	111	28	83	73
30.0	0.301	81	24	56	84	97	29	68	70
32.5	0.325	80	26	54	79	97	31	65	65
35.0	0.349	69	24	45	85	84	29	55	70
37.5	0.373	71	26	44	78	85	32	53	65

**Table 3 molecules-26-03946-t003:** Calculated values of the bilayer volume fraction (Φ_bi_), bilayer thickness (d_bi_) and area per surfactant molecule (a) of the 25 wt% of the DODAB and 35 wt% of the DODAC bilayers in water at temperatures below and above the main phase transition temperature (T_m_).

		Φ_bi_	q (Å^−1^)	d_sp_ (Å)	d_bi_ (Å)	a (Å^2^)
<T_m_	DODAB	0.240	0.174	36	9	237
	DODAC	0.349	0.093	69	24	85
>T_m_	DODAB	0.240	0.050	126	30	68
	DODAC	0.349	0.075	84	29	70

**Table 4 molecules-26-03946-t004:** Normalised differential scanning calorimetry parameters of 35 wt% DODAB and 35 wt% DODAC in the presence of 0.25 wt% and 0.50 wt% NaBr and NaCl, respectively.

Surfactant	Salt (wt%)	Heating Cycle		Cooling Cycle	
		T_m_ (°C)	ΔH_m_ (kJ mol^−1^)	T_c_ (°C)	ΔH_c_ (kJ mol^−1^)
35 wt% DODAB	0.00	53.8	108.8	47.1	100.4
35 wt% DODAB	0.25	54.0	95.6	43.3	103.5
35 wt% DODAB	0.50	54.0	101.7	45.2	106.9
35 wt% DODAC	0.00	44.9	44.6	43.7	49.9
35 wt% DODAC	0.25	44.8	40.2	43.6	43.6
35 wt% DODAC	0.50	44.4	42.3	43.3	48.9

**Table 5 molecules-26-03946-t005:** Values of the interlamellar spacing (d_sp_) of 35 wt% DODAB and 35 wt% DODAC bilayers in the presence of 0.25 wt% and 0.50 wt% NaBr and NaCl, respectively. The measurements were performed at temperatures below (25 °C) and above (65 °C) the main phase transition temperature.

	Surfactant (wt%)	Salt (wt%)	d_sp_, 25 °C (Å)	d_sp_, 65 °C (Å)
DODAB	35.0	0.00	36	126
		0.25	36	122
		0.50	36	96
DODAC	35.0	0.00	69	84
		0.25	75	95
		0.50	66	81

**Table 6 molecules-26-03946-t006:** Logarithm of the octanol/water partition coefficient for additives investigated.

Compound	Log P [47]
Water	-
DODAB	3.80
DODAC	3.80
Urea	−2.11
Methyl urea	−1.40
Dimethyl urea	−0.49
Acetic acid	−0.17
Propionic acid	0.33
Butyric acid	0.79
Sodium butyrate	0.79
Hexanoic acid	1.92
Benzyl alcohol	1.05
Phenoxyethanol	1.13
1-Butanol	0.84
1-Hexanol	2.03
1-Octanol	3.07
1-Decanol	4.57
1-Dodecanol	5.13

**Table 7 molecules-26-03946-t007:** Unit cell values calculated from the X-ray scattering patterns of DODAC in the presence of decanol and dodecanol at 65 ^o^C for a reversed hexagonal phase. r_w_ is the polar domain radius.

Additive	Additive (wt%)	Above T_m_			Transition
		d_100_ (Å)	b (Å)	r_w_ (Å)	
Decanol	15.0	48	56	20	L_β—_rev Hex
	20.0	45	52	20	L_β—_rev Hex
Dodecanol	12.5	54	62	22	L_β—_rev Hex
	15.0	51	58	22	L_β—_rev Hex
	20.0	49	56	22	L_β—_rev Hex

## Data Availability

Not applicable.

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
