# Peer review of "Double-Chain Cationic Surfactants: Swelling, Structure, Phase Transitions and Additive Effects"

_molecules, 2021, doi:10.3390/molecules26133946_

Round 1

Reviewer 1 Report

The present paper of Lindman and coworkers discusses an important topic in surfactant science, the phase behavior of archetype double chain cationic surfactant systems, used extensively in cosmetic formulations. Authors present plenty of experimental data, accompanied with a thorough interpretation and provide valuable data and information, which are certainly worth of publication.

The paper is well written, and the introduction is especially very pleasant to read.

I have one remark. Authors interpret the high temperature phase of water – DODAC – butanol system as sponge phase, based on the observation of a microemulsion like scattering pattern. However, neither in the data, nor in the cited literature I cannot see any strong evidence on the formation of the sponge phase. The TS like behavior is not a signature of a sponge phase.

I suggest, authors reconsider their interpretation, in order to prevent the further spread of a potentially misleading information. Or, support their view by a clear discussion and strong argumentation.

Interestingly, a close look at the scattering pattern may hint to the presence of strongly smeared reflections of a hexagonal structure. A repeated, very accurate measurement of this system with some variations of the alcohol content should be very informative, and if the time allows, I would recommend to verify these observations and improve this part of the paper.

in ref. 46, the chapter number 15 is probably not needed.

Author Response

Dear Reviewer,

Thank you for your valuable comments.

Please find the answers to your remarks in the document attached.

Kind regards,

Rui A. Gonçalves

on behalf of my co-workers.

Reviewer 2 Report

In this work, self-assembly in two double-chain cationic surfactant systems, dioctadecyldimethylammonium chloride (DODAC) and dioctadecyldimethylammonium bromide (DODAB) in water, was investigated by means of differential scanning calorimetry and small- and wide-angle X-ray scattering. Thermodynamic parameters of the first-order Lβ-Lα phase transition and the characteristics of the two lamellar phases were obtained for different surfactant volume fractions. The effect of salts and other additives was investigated. The manuscript contains some important experimental findings. In particular, from the bilayer spacing in Lβ phase it was concluded that a structure with surfactant molecules being oriented perpendicularly to the bilayer is not realized under the conditions of this study. I think the paper is sound and certain results could be interesting for experimentalists. There are, however, some minor points, which should be considered before the publication.

Lines 259-260: "To analyze the SAXS and WAXS results, the values used for the various parameters were: ?ℎ?,? = 1026 Å3 [36], ?? = 1126 Å3 [37], and ?? = 30 Å3 [35]." - Can these parameters be considered as constants, independent of the experimental conditions?

Lines 327-329: A "pre-transition" should be explained in more details.

Lines 358-360: "This is in line with the expected decrease from a “solid-like” to “fluid-like” hydrocarbon state, assuming all the chains are in a trans conformation and are perpendicular to the interface in the Lβ phase [46,47]." - This statement is not quite clear.

Lines 368-369: "The Tm remained unchanged for both surfactants, but a slight decrease of the enthalpy of phase transition on melting was detected." - There is a slight increase of the enthalpy in the cooling cycle for DODAB. Does the variation of Tm or enthalpy exceed the experimental error? How large were experimental errors in these and other experiments?

Lines 375-377: "NaCl screens the headgroup charges, which resulted in a closer packing of the surfactant molecules, hence a further swelling to 95 Å was recorded." - The further increase of salt concentration results in a decrease of dsp to 81 Å. The variation of some characteristics with addition of salts in Tables 4 and 5 looks like non-monotonic or scattering. Does this mean that the salt effect is not significant within the experimental error?

Table 5: Was the DODAB concentration 35 wt% or 25 wt% ? Was the temperature above the main phase transition for DODAC 65 oC or 50 oC ?

Lines 609-610: "It would, therefore, be of interest to have thermodynamically stable systems of Lα, which on dilution spontaneously transform into the Lβ phase." - The authors do not discuss the temperature range where such transformation occurs. Is it possible to do such transformation, e.g., at room temperatures? According to the data in Table S3, Tm = 19.1 oC for 12.5 wt% Octanol system. But, in a diluted system Tm increases. Could the authors clarify the role of temperature?

Author Response

(The authors gave the same response as above.)

Reviewer 3 Report

The authors have carried out an extensive experimental study of the phase diagrams of two double-chain cationic surfactants with different chain lengths. The experimental techniques used are DSC, SAXS and WAXS, that have allow the authors to discuss the structure and the chain orientation in the La and Lb phases. The most important results are related to the effect of many additives on the phase diagram. The manuscript does not represent any breakthrough in the current knowledge of the phase diagrams, and needs some changes before it can be accepted for publication:

COMMENTS

1.- The authors have used, among others, long-chain alcohols as additives. It is well known (e.g. see J. Venzmer) that mixtures of alcohols with different chain lengths  form lamellar phases. I wonder whether it could happen that the alcohols might interdigitate with the chains of the surfactants, thus changing the structure and properties of the system. The authors should mention, and if possible discard, this possibility.

2.- The authors should mention the precision of the experimental data and give an estimation of the uncertainty of their results, e.g. bilayer thickness, etc.

3.- The mobility/rigidity of the different lamellar phases could be easily tested y measuring their microviscosity using the ESR technique (see Mateos-Maroto et al. Langmuir (2021). This possibility should, at least, be mentioned because it is very sensitive to the order-disorder transitions and can be easily done as a function of T, thus allowing a very complementary information to the X-ray results.

Author Response

(The authors gave the same response as above.)
